# Role of Erythropoietin Receptor Signaling in Macrophages or Choroidal Endothelial Cells in Choroidal Neovascularization

**DOI:** 10.3390/biomedicines10071655

**Published:** 2022-07-09

**Authors:** Aniket Ramshekar, Colin A. Bretz, Eric Kunz, Thaonhi Cung, Burt T. Richards, Gregory J. Stoddard, Gregory S. Hageman, Brahim Chaqour, M. Elizabeth Hartnett

**Affiliations:** 1Department of Ophthalmology and Visual Sciences, John A. Moran Eye Center, University of Utah, 65 Mario Capecchi Dr, Salt Lake City, UT 84132, USA; u1088294@utah.edu (A.R.); u6003023@utah.edu (C.A.B.); eric.kunz@utah.edu (E.K.); u6036884@utah.edu (T.C.); 2Sharon Eccles Steele Center for Translational Medicine, John A. Moran Eye Center, University of Utah, 65 Mario Capecchi Dr, Salt Lake City, UT 84132, USA; burt.richards@hsc.utah.edu (B.T.R.); gregory.hageman@hsc.utah.edu (G.S.H.); 3Department of Internal Medicine, University of Utah, 30 N 1900 E, Salt Lake City, UT 84132, USA; greg.stoddard@hsc.utah.edu; 4Department of Ophthalmology, University of Pennsylvania, 422 Curie Boulevard, Philadelphia, PA 19104, USA; brahim.chaqour@pennmedicine.upenn.edu

**Keywords:** EPOR, EPO, CNV, AMD, CECs, macrophages

## Abstract

Erythropoietin (EPO) has been proposed to reduce the progression of atrophic age-related macular degeneration (AMD) due to its potential role in neuroprotection. However, overactive EPO receptor (EPOR) signaling increased laser-induced choroidal neovascularization (CNV) and choroidal macrophage number in non-lasered mice, which raised the question of whether EPOR signaling increased CNV through the recruitment of macrophages to the choroid that released pro-angiogenic factors or through direct angiogenic effects on endothelial cells. In this study, we addressed the hypothesis that EPOR signaling increased CNV by direct effects on macrophages or endothelial cells. We used tamoxifen-inducible macrophage-specific or endothelial cell-specific EPOR knockout mice in the laser-induced CNV model, and cultured choroidal endothelial cells isolated from adult human donors. We found that macrophage-specific knockout of EPOR influenced laser-induced CNV in females only, whereas endothelial-specific knockout of EPOR reduced laser-induced CNV in male mice only. In cultured human choroidal endothelial cells, knockdown of EPOR reduced EPO-induced signal transducer and activator of transcription 3 (STAT3) activation. Taken together, our findings suggest that EPOR signaling in macrophages or choroidal endothelial cells regulates the development of CNV in a sex-dependent manner. Further studies regarding the role of EPO-induced signaling are required to assess EPO safety and to select or develop appropriate therapeutic approaches.

## 1. Introduction

Age-related macular degeneration (AMD) is a leading cause of blindness worldwide [1]. The most severe vision loss is found in the advanced forms of the disease, atrophic and neovascular AMD [2,3]. There is currently no highly efficacious approved treatment for atrophic AMD, despite the involvement of complement, inflammatory factors, and oxidative stress, whereas agents that interfere with angiogenic factors (e.g., vascular endothelial growth factor [VEGF]) are used for neovascular AMD to reduce vision-threatening macular or choroidal neovascularization (CNV) [4,5,6]. Erythropoietin (EPO), a hematopoietic hormone, has been proposed as a potential therapeutic target to reduce the progression of atrophic AMD due to its role as a neurotrophic factor in the retina [7,8]. However, elderly patients with active CNV had increased serum EPO levels compared to patients with inactive CNV [9]. This finding was supported in a translational neovascular AMD model, the murine laser-induced CNV model, where we previously found C57Bl/6J mice had increased serum EPO associated with increased laser-induced CNV [10]. These findings suggest that EPO may be involved in the progression of neovascular AMD and raise concerns regarding the use of EPO in atrophic AMD. Understanding the role of EPO-triggered signaling in CNV is warranted to assess the safety of EPO use in elderly AMD patients and to select effective and safe therapeutic strategies.

Studies have demonstrated that EPO triggers signaling through its receptor, EPO receptor (EPOR), to mediate non-hematopoietic effects involving angiogenesis or tissue protection [11,12]. EPOR has been identified in several cell types within the retina, including the retinal pigment epithelium (RPE) and neurons, supporting the thinking that EPO-induced EPOR signaling might confer neuroprotection [13,14,15,16]. One challenge in assessing EPO signaling through EPOR is that *Epo* or *Epor* knockout is embryonic lethal in mice [17,18]. To address this limitation, we previously used transgenic knock-in mice that expressed either a humanized EPOR that reduced EPOR signaling, or a mutant humanized *EPOR* gene that overactivated EPOR signaling and compared outcomes to control littermate wildtype mice [19]. We found that signaling through EPOR was important in vascular repair, angiogenesis and neuroprotection in the mouse retina and oxygen-induced retinopathy model [10,20,21]. In addition, compared to littermate control mice, overactive EPOR signaling in the transgenic knock-in mice was associated with an increased number of choroidal macrophages before laser injury and significantly increased experimental CNV one week after laser injury [10]. These observations led us to postulate that EPOR-mediated signaling increases the development of macular or choroidal neovascularization by direct effects on macrophages or choroidal endothelial cells. The current study addresses this hypothesis by using tamoxifen-inducible cell-specific EPOR knockout mice in the laser-induced CNV model.

## 2. Materials and Methods

### 2.1. Animals and Ethical Statement

All animal procedures were performed according to protocols that were approved by the Institutional Animal Care and Use Committee (IACUC) and the Institutional Biosafety Committee at the University of Utah, and in accordance with the University of Utah (Guide for the Care and Use of Laboratory Animals) and the Association for Research in Vision and Ophthalmology Statement for the Use of Animals in Ophthalmic and Vision Research. Both male and female mice were used in all experiments. Experimental mice were on a C57/Bl6J genetic background and negative for *Rd1*, *Rd8*, and *Gnat2* mutations [22].

### 2.2. Generation of Conditional Knockout Mice and Tamoxifen Injections

*Epor*^tm1A(KOMP)Wtsi^ mice (KOMP, University of California, Davis, CA, USA) were crossed with *Rosa26*^tm1(FLP1)Dym^ mice (The Jackson Laboratory, Bar Harbor, ME, USA) to ultimately generate *Epor*^flox/flox^ mice. Offspring from *Cdh5*-CreERT2^+/+^ or *Cx3cr1*-CreERT2^+/+^ (The Jackson Laboratory, Bar Harbor, ME, USA) crossed with *Rosa26*-tdTomato^flox/flox^ (The Jackson Laboratory) were bred with *Epor*^flox/flox^ mice to ultimately generate *Cdh5*-CreERT2^+/−^; *Rosa26*-tdTomato^flox/flox^; *Epor*^flox/flox^ (EPOR^iΔEC^) or *Cx3cr1*-CreERT2^+/−^; *Rosa26*-tdTomato^flox/flox^; *Epor*^flox/flox^ (EPOR^iΔMΦ^) mice, respectively, along with littermate control mice that had no copies of either Cre allele (EPOR^fl^). EPOR^iΔEC^, EPOR^iΔMΦ^, and littermate EPOR^fl^ mice between four and six weeks of age received intraperitoneal tamoxifen reconstituted in corn oil once every other day (2 mg/day) for three injections.

### 2.3. Laser-Induced Choroidal Neovascularization Model

The laser-induced CNV model was performed following a standardized protocol two weeks after tamoxifen administration [23]. Experimental mice were weighed and anesthetized with intraperitoneal ketamine (100 mg/kg, California Pet Pharmacy, Hayward, CA, USA) and xylazine (10 mg/kg, Akorn, Lake Forest, IL, USA). Once unresponsive to a toe-pinch with observable steady breathing, mice were administered one drop of proparacaine hydrochloride ophthalmic solution (Bausch & Lomb Inc., Laval, QC, Canada) and tropicamide ophthalmic solution (Akorn, Lake Forest, IL, USA) in both eyes. Mice were positioned on a platform in front of the Micron IV Image-Guided Laser System (Phoenix Research Laboratories, Pleasanton, CA, USA) and a coupling agent, GenTeal (Alcon, Geneva, Switzerland), was applied to the corneas of both eyes. Retinal fluorescence was assessed in all mice before laser-induced injury. Mice received three laser-induced burns (450 mW intensity, 100 ms duration) per eye roughly two-disc diameters from the optic nerve while avoiding major blood vessels. Successful laser-induced injury was confirmed by the appearance of a cavitation bubble, indicating the disruption of Bruch’s membrane, which lacked bleeding from nearby blood vessels. After laser-induced injury, mice were placed on a pre-warmed heating pad and were closely monitored until they fully recovered from anesthesia. One week after laser-induced injury, mice were euthanized, eyes were collected for flat mounts or RT-PCR analysis, and tissues were collected for RT-PCR analysis. 

### 2.4. Posterior Eye Cup Flat Mounts and Immunostaining

Immediately after enucleation of the eye, corneas were clipped, fixed in 4% paraformaldehyde (Electron Microscopy Sciences, Hatfield, PA, USA) diluted in 1× phosphate buffered saline (PBS, Thermo Fisher Scientific, Waltham, MA, USA) for one hour, and subsequently washed three times in 1× PBS. After removing the extraocular muscles, corneas, and lenses, the posterior eye cups consisting of the RPE/choroid/sclera were separated from the retina and incubated in 1× PBS containing 5% normal goat serum and 0.4% TritonX-100 (i.e., blocking solution) for one hour at room temperature. Subsequently, the posterior eye cups were incubated in blocking solution supplemented with AlexaFluor 647-conjugated Isolectin B4 antibodies (1:500, Invitrogen, Carlsbad, CA, USA) and rabbit anti-red fluorescent protein antibodies (anti-RFP, 1:200, Abcam, Waltham, MA, USA) overnight at 4 °C. Posterior eye cups were washed three times in 1× PBS and incubated with blocking solution supplemented with Cy3-conjugated goat anti-rabbit (1:500, Jackson ImmunoResearch Inc., West Grove, PA, USA) for three hours at room temperature. Posterior eye cups were washed three times in 1× PBS and radial incisions, avoiding laser lesions, were performed to flat mount RPE/choroids onto a microscope slide with a Vectashield mounting medium (Vector Laboratories, Burlingame, CA, USA).

### 2.5. Imaging and Quantifying Laser-Induced Lesion Volumes

Confocal Z-stack images of each lesion were acquired using a Confocal Laser Scanning Microscope (Olympus Corporation, Tokyo, Japan) at 20× objective. Confocal images of each lesion were uploaded into IMARIS (Bitplane, Zurich, Switzerland). Lesion volumes were measured using the surfaces feature in IMARIS and manually verified by a masked reviewer, as previously described [10]. CNV lesions from eyes with bleeds from either laser-induced injury or bridging CNV lesions were excluded from statistical analysis, as per standard protocol [23].

### 2.6. Choroidal Endothelial Cell Culture Conditions

CECs were isolated from de-identified human eyes from adult donors less than 50 years of age, obtained from the Utah Lions Eye Bank (Salt Lake City, UT, USA), and procured and processed by the Steele Center for Translational Medicine, as previously described [24]. Genotyping was performed on donor cells and CECs isolated from donors with low to moderate risk of AMD were used for experiments. CECs, passages 3–5, were cultured in attachment factor (Cell Systems, Kirkland, WA, USA)-coated cultureware and Endothelial Growth Medium-2 BulletKit supplemented with 5% fetal bovine serum (EGM-2, Lonza, Walkersville, MD, USA) as the growth media.

### 2.7. Transfection and Treatment

CECs were transfected with an equal concentration of either control small interfering RNA (siRNA) or EPOR siRNA oligos using Lipofectamine 3000 following the protocol recommended by the manufacturer (Life Technologies, Grand Island, NY, USA). Forty-eight hours after transfection, CECs were serum-starved in Endothelial Basal Media-2 (EBM-2, Lonza, Walkersville, MD, USA) for three hours and then treated with either recombinant human VEGF (25 ng/mL, R&D Systems, Minneapolis, MN, USA), recombinant human EPO (5 U/mL, R&D Systems, Minneapolis, MN, USA) or PBS as vehicle control for 30 min. All treatments were volume-controlled.

### 2.8. RNA Extraction and Quantitative RT-PCR Analysis

Immediately following treatment, cultured CECs in cultureware plates were washed once with ice-cold 1× PBS and lysed in buffer RLT supplemented with β-mercaptoethanol (1:100, Sigma-Aldrich, St. Louis, MO, USA). Lysates were collected in 1.5 mL microcentrifuge Eppendorf tubes, and RNA was extracted from lysates using the RNeasy Kit following the protocol provided by the manufacturer (Qiagen, Valencia, CA, USA).

Immediately after being collected from euthanized mice, tissues were snap-frozen in liquid nitrogen. Snap-frozen tissues were stored at −80 °C. For RT-PCR, tissues were placed and sonicated in buffer RLT supplemented with β-mercaptoethanol (1:100, Sigma-Aldrich, St. Louis, MO, USA), and RNA was extracted from each sample using the RNeasy Kit following the protocol provided by the manufacturer (Qiagen, Valencia, CA, USA).

Isolated RNA samples were reverse transcribed to cDNA using the High Capacity cDNA Reverse Transcription Kit (Thermo Fisher Scientific, Waltham, MA, USA). The cDNA of each sample was evaluated for the expression of specific genes using the TaqMan Gene Expression Master Mix and TaqMan probes against the genes of interest (Thermo Fisher Scientific, Waltham, MA, USA, Appendix A). The ΔCT values for the specific genes were calculated using β-actin as the house-keeping gene control, and the 2^−ΔΔCT^ was calculated relative to the experimental control group.

### 2.9. Protein Extraction and Western Blot Analysis

Following treatment, cultured CECs in cultureware plates were washed once with ice-cold 1× PBS and lysed in radioimmunoprecipitation assay (RIPA) buffer supplemented with 1X phosphatase (Thermo Fisher Scientific, Waltham, MA, USA) and 1× protease (Millipore Sigma, Burlington, MA, USA) inhibitors. Lysate samples were collected in 1.5 mL microcentrifuge Eppendorf tubes and centrifuged for 5 min at 16,000× *g* at 4 °C. Supernatants were collected and a bicinchoninic acid assay was performed to determine protein concentrations. An equal concentration of protein from each sample was suspended in 1× sample buffer (Thermo Fisher Scientific, Waltham, MA, USA) and denatured at 95 °C for 5 min.

An equal volume of the denatured protein from each sample was loaded into wells of a NuPAGE 4–12% Bis-Tris Gel (Thermo Fisher Scientific, Waltham, MA, USA) and electrophoresis was performed at 80 V for 3 h. Separated proteins in the gel were transferred to a membrane in the iBlot 2 PVDF transfer stack (Thermo Fisher Scientific, Waltham, MA, USA) using the iBlot 2 Dry Blotting System (Thermo Fisher Scientific, Waltham, MA, USA). Membranes were briefly rinsed in 1× Tris-buffered saline (TBS, Quality Biological, Gaithersburg, MD, USA) supplemented with 0.1% Tween-20 (TBST, Cole-Parmer, Vernon Hills, IL, USA) and then blocked for one hour at room temperature in 5% bovine serum albumin (BSA, Sigma-Aldrich, St. Louis, MO, USA) in 1× TBS. Membranes were incubated in 5% BSA in 1× TBS supplemented with rabbit anti-phospho-STAT3 (Y705, 1:1000, Cell Signaling, Danvers, MA, USA) or mouse anti-STAT3 (1:1000, Cell Signaling, Danvers, MA, USA) overnight at 4 °C. Membranes were washed three times in TBST and incubated in 5% BSA in 1× TBS supplemented with either horseradish peroxidase (HRP)-conjugated goat anti-rabbit (1:3000, Thermo Fisher Scientific, Waltham, MA, USA) or HRP-conjugated mouse anti-mouse (1:3000, Cell Signaling, Danvers, MA, USA) for one hour at room temperature. Membranes were washed three times in TBST, developed in chemiluminescent HRP substrates (Thermo Fisher Scientific, Waltham, MA, USA), and imaged using the C-DiGit Blot Scanner (LI-COR Biotechnology, Lincoln, NE, USA) and Image Studio software (v5.2, LI-COR Biotechnology, Lincoln, NE, USA). Membranes were re-probed with HRP-conjugated mouse anti-β-actin (1:3000, Santa Cruz Biotechnology, Dallas, TX, USA) as loading control in 5% milk in 1× TBS and re-imaged as described above. Densitometry quantification was performed using Fiji software (National Institutes of Health, Bethesda, MD, USA) [25].

### 2.10. Statistical Analyses

All statistical analyses were performed in Stata-17 (StataCorp LLC, College Station, TX, USA). For the in vivo laser-induced CNV study, the data were analyzed with a mixed effects linear regression model with lesions nested within the same eye using CNV volume measurements as the continuous outcome variable. For the in vivo RT-PCR study, the data were analyzed with a mixed effects linear regression model with mice nested within a litter using the Δ*C*_T_ as the continuous outcome variable. For these two studies, we used treatment, sex, and treatment × sex interactions as the predictor variables. When there is an interaction term in the mixed effects model, the main effect terms cannot be interpreted. Therefore, we estimated the main effect terms and adjusted means using post-fit marginal estimation with a Wald test for significance testing.

For in vitro RT-PCR study, the data were analyzed with a mixed effects linear regression model treating CECs isolated from different human donors as a random effect using the Δ*C*_T_ as the continuous outcome variable. For the in vitro western blot study, the data were analyzed with a mixed effects linear regression model treating CECs isolated from different human donors as a random effect using densitometry measurements as the continuous outcome variable. In both models, the only predictor variable was treatment. We did not test for sex effects in our in vitro studies because we only had two male donors and one female donor and, therefore, did not have sufficient biologic variation to make reliable sex inferences. Results are presented as mean ± standard error (SE) and a *p*-value < 0.05 was considered statistically significant.

## 3. Results

### 3.1. EPOR Expression in RPE/Choroids from Human Donor Eyes

In order to confirm the expression of *EPO* and *EPOR* mRNA in RPE/choroids from human donor eyes, we assessed expression of the mRNAs in RPE/choroids from macular or extramacular specimens from human donors with or without AMD that were collected in a previous study [26]. *EPO* and *EPOR* mRNAs were expressed in all RPE/choroid samples without differences among the groups (Figure 1A,B).

Several cell populations are present within human RPE/choroid tissues [27]. We, therefore, wanted to confirm the expression of *EPOR* mRNA in specific cell populations. To address this consideration, we analyzed single-cell RNA-sequencing data from macular RPE/choroids of donors with AMD or donors without AMD as controls from a publicly available website (https://singlecell-eye.org (accessed on 12 April 2022)) and publication [28,29]. *EPOR* mRNA was expressed in macrophages (e.g., infiltrative and resident) and choriocapillaris cells isolated from all donors included in the study (Figure 1C). Taken together, the data suggest that EPOR is expressed in macrophages and endothelial cells in RPE/choroids isolated from donors with and without AMD.

**Figure 1 biomedicines-10-01655-f001:**
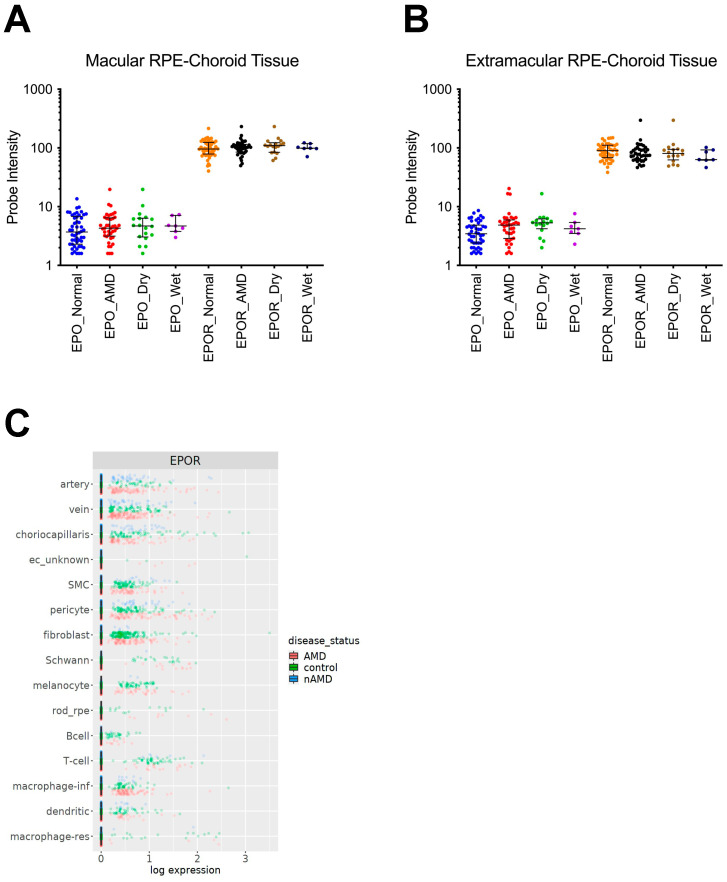
*EPOR* mRNA expression in RPE/choroids from adult human donor eyes. *EPO* mRNA was detected in (**A**) macular and (**B**) extramacular RPE/choroids that were isolated from donors with AMD (red dots), including dry AMD (green dots) and wet AMD (purple dots), or without AMD (blue dots), and *EPOR* mRNA was detected in (**A**) macular or (**B**) extramacular RPE/choroids that were isolated from donors with AMD (black dots), including dry AMD (brown dots) and wet AMD (dark blue dots), or without AMD (orange dots) that were collected in a previous publication [26]; (**C**) *EPOR* mRNA was also detected in single cells dissociated from RPE/choroids from patients without AMD (green dots) or with dry AMD (red dots) or wet AMD (blue dots) that were collected in a previous publication [28,29].

### 3.2. The Effect of EPOR-Triggered Signaling in Macrophages on Laser-Induced Choroidal Neovascularization

We previously observed increased choroidal macrophages in mice with overactive EPOR signaling compared to control littermate wildtype mice before laser-induced injury, and greater lectin-stained CNV volume compared to littermate wildtype mice one week after laser-induced injury [10]. These findings led us to postulate that EPOR signaling in macrophages is necessary for the development of CNV. To test this hypothesis, we used the tamoxifen-inducible Cre-loxP knockout mouse model to generate macrophage-specific EPOR knockout mice (EPOR^iΔMΦ^) and littermate control mice that lack copies of the Cre allele (EPOR^fl^, Figure 2A). We observed diffuse and speckled tdTomato fluorescence in the retina by live Micron IV imaging (Figure 2B) and by confocal microscopy of fixed retinal flat mounts (Appendix A) in tamoxifen-injected EPOR^iΔMΦ^, but not in tamoxifen-injected littermate EPOR^fl^. We also observed reduced *Epor* mRNA expression in splenic tissues from tamoxifen-injected EPOR^iΔMΦ^ compared to tamoxifen-injected littermate EPOR^fl^ (Appendix A). These findings confirmed Cre-mediated recombination of the tdTomato reporter and *Epor* gene in our transgenic mouse model.

We then performed laser-directed injury to induce CNV in the transgenic mice and did not observe a significant difference in lectin-stained CNV volume in tamoxifen-injected EPOR^iΔMΦ^ compared to tamoxifen-injected EPOR^fl^ (Figure 3A,B). We stratified the CNV volume data by sex and observed significantly increased CNV volume in EPOR^iΔMΦ^ compared to littermate EPOR^fl^, but only in female mice (Figure 3C). Taken together, our findings suggest that EPOR-triggered signaling in macrophages is involved in the development of CNV, but only in female mice.

There was no significant difference in lectin-stained CNV volume in tamoxifen-injected male EPOR^fl^ compared to tamoxifen-injected female EPOR^fl^ (*p*-value = 0.78). 

### 3.3. The Effect of EPOR-Triggered Signaling in Endothelial Cells on Laser-Induced Choroidal Neovascularization

In our previous study, compared to respective control littermate wildtype mice, we observed increased laser-induced CNV volume in mice with overactive EPOR signaling, and a trend toward reduced laser-induced CNV volume in mice with hypoactive EPOR signaling [10]. These findings suggested that EPOR signaling in endothelial cells might be involved in the development of CNV. To determine if endothelial EPOR-triggered signaling is important in the development of CNV, we used the tamoxifen-inducible Cre-loxP knockout mouse model to generate endothelial-specific EPOR knockout mice (EPOR^iΔEC^) and littermate control mice that lack copies of the Cre allele (EPOR^fl^, Figure 4A). We confirmed Cre-mediated recombination of the tdTomato reporter by observing tdTomato fluorescence lining the retinal vessels by live Micron IV imaging (Figure 4B) and confocal microscopy of fixed retinal flat mounts (Appendix A) in tamoxifen-injected EPOR^iΔEC^ but not in tamoxifen-injected littermate EPOR^fl^.

We then used the tamoxifen-injected EPOR^iΔEC^ and littermate EPOR^fl^ in the laser-induced CNV model. Compared to tamoxifen-injected littermate EPOR^fl^, we observed no significant changes in CNV volume in the tamoxifen-injected EPOR^iΔEC^ (Figure 5A,B). We stratified the CNV volume by sex and observed a significant ~50% reduction in tamoxifen-injected male EPOR^iΔEC^ compared to tamoxifen-injected littermate male EPOR^fl^; however, we did not observe a significant difference in CNV volume between the tamoxifen-injected female EPOR^iΔEC^ compared to the tamoxifen-injected littermate female EPOR^fl^ (Figure 5C). Taken together, the data suggest that EPOR signaling in endothelial cells is necessary for the development of CNV in male mice and that endothelial EPOR signaling is involved.

There was no significant difference in lectin-stained CNV volume in tamoxifen-injected male EPOR^fl^ compared to tamoxifen-injected female EPOR^fl^ (*p*-value = 0.28). 

### 3.4. The Effect of EPOR-Triggered Signaling in Choroidal Endothelial Cells

A previous study reported that EPOR-mediated signaling was not detected in several cultured endothelial cells [30]. However, we previously found that EPOR-triggered signaling was necessary for VEGF-induced STAT3 activation in cultured human retinal endothelial cells (HRMECs) [31]. Furthermore, STAT3 activation was detected in CNV lesions of AMD patients [32]. Therefore, we sought to determine if EPOR-triggered STAT3 activation occurs in cultured CECs. To validate sufficient EPOR knockdown in cultured CECs, we assessed *EPOR* mRNA by RT-PCR. Forty-eight hours after transfection, cultured CECs transfected with EPOR siRNA had a significant 70% reduction in EPOR mRNA expression compared to cultured CECs transfected with control siRNA (Figure 6A). Compared to control siRNA-transfection, EPOR siRNA-transfection was associated with significantly reduced STAT3 activation by VEGF or EPO in cultured CECs (Figure 6B,C). In addition, knockdown of EPOR did not reduce KDR mRNA expression or VEGFR2 protein expression in cultured CECs, but rather increased mRNA and protein expression (data not shown). Taken together, these findings suggest that EPOR-triggered signaling is necessary for STAT3 activation by angiogenic factors implicated in AMD pathology.

## 4. Discussion

Severe vision loss in AMD is found in the advanced stages, atrophic and neovascular AMD. Although neovascular AMD can be treated with anti-angiogenic agents, they are effective in only ~50% of patients [6,33,34,35,36]. Also, there are currently no FDA-approved treatments available for atrophic AMD. Therefore, there is a need to identify alternative therapeutic approaches. Systemic administration of EPO is used to treat adult anemia and has been discussed as a potential treatment for atrophic AMD due to its neuroprotective effects on the retina [7,37]; however, there are associations between increased serum EPO and severity of CNV in AMD patients [9]. Furthermore, EPO-induced neural protection and angiogenesis have been demonstrated in experimental models through EPOR-mediated signaling [10,21]. Therefore, we predicted that EPOR signaling might be involved in the development of CNV.

We found that *EPOR* mRNA is expressed in several cell types in RPE/choroids isolated from human donors, including macrophages and choriocapillaris endothelial cells using a publicly available database [29]. We previously observed that the RPE/choroids of mice with overactive EPOR signaling had more macrophages prior to laser-induced injury, greater concentrations of inflammatory cytokines three days after laser-induced injury, and increased laser-induced CNV seven days after laser-induced injury compared to littermate wildtype control mice [10]. These data suggested that EPOR-mediated signaling might also be involved in the recruitment of choroidal macrophages that release inflammatory/angiogenic cytokines that influence the development of CNV. In support of this thinking, EPOR expression has been demonstrated in macrophages and EPOR-mediated release of angiogenic or inflammatory cytokines has been demonstrated in peripheral blood mononuclear cells [38,39,40,41,42]. We, therefore, postulated that EPOR signaling in choroidal macrophages would promote the development of CNV. However, we observed no significant difference in CNV volume between tamoxifen-injected EPOR^iΔMΦ^ and EPOR^fl^. When we stratified the data by sex, however, we unexpectedly found that tamoxifen-injected female EPOR^iΔMΦ^ had significantly increased laser-induced CNV volume compared to littermate tamoxifen-injected female control mice. There is limited literature to support sex differences in macrophage-specific EPOR-mediated signaling. Nonetheless, our findings suggest that EPOR signaling in macrophages is involved in the development of CNV in a sex-dependent manner.

EPOR is also expressed in endothelial cells and EPOR signaling has been associated with angiogenesis [10,31,43]. We, therefore, tested the hypothesis that endothelial EPOR is necessary for the development of CNV. We did not observe a significant difference in CNV volume between tamoxifen-injected EPOR^iΔEC^ and tamoxifen-injected EPOR^fl^. When we stratified the data by sex, however, we found significantly reduced CNV volume in tamoxifen-injected male EPOR^iΔEC^ compared to littermate tamoxifen-injected male EPOR^fl^, but not in respective female groups. Interestingly, endothelial-specific EPOR knockout resulted in significant differences in the male mice, whereas macrophage-specific EPOR knockout resulted in significant differences in the female mice. We previously did not observe sex-related differences in mice with hypoactive or overactive EPOR signaling compared to littermate control wildtype mice [10]. However, we have observed significant differences in CNV volume in tamoxifen-injected female non-Cre control mice compared to tamoxifen-injected male non-Cre control mice, even though tamoxifen-induced Cre-mediated recombination reduced CNV in male mice compared to tamoxifen-injected littermate male controls [24]. These findings suggest that the sex-related differences observed in the transgenic EPOR knockout mouse models might be due to an interaction between CNV and tamoxifen administration. Therefore, further studies in the tamoxifen-inducible Cre-loxP mouse model are warranted to determine the effects of tamoxifen on laser-induced CNV. Nonetheless, the data from male mice in the endothelial EPOR knockout model support our hypothesis and suggest that endothelial EPOR signaling might be necessary for the development of CNV.

To corroborate our findings from the male endothelial-specific EPOR knockout mice, we sought to determine the signaling effects of EPOR knockdown in cultured CECs. There are several EPOR-mediated downstream effectors in endothelial cells (e.g., STATs, PI3K/AKT, ERK) [44]. In HRMECs, knockdown of EPOR reduced STAT3 activation by VEGF [31]. Furthermore, STAT3 expression was detected in CNV lesions from human donors, and preventing STAT3 activation with a pharmacologic JAK2 inhibitor significantly reduced laser-induced CNV compared to vehicle control in C57/Bl6J mice [32,45]. We therefore predicted that EPOR is necessary for STAT3 activation in cultured CECs isolated from male and female donor eyes with low to moderate AMD risk. In support of our prediction, we observed that EPOR knockdown significantly reduced STAT3 activation by VEGF or EPO despite increased VEGFR2 expression, which might be a compensatory response. Taken together, our findings suggest EPOR-mediated STAT3 activation in CECs is important for the development of CNV.

In conclusion, our findings suggest that endothelial or macrophage EPOR-mediated signaling regulates the development of CNV in a sex-dependent manner. A previous study found that overexpression of a modified form of EPO specifically in the RPE reduced retinal degeneration in response to oxidative stress [46]. Therefore, further studies regarding the role of EPO in AMD as well as the role of EPO-mediated signaling in different cell types are warranted.

## Figures and Tables

**Figure 2 biomedicines-10-01655-f002:**
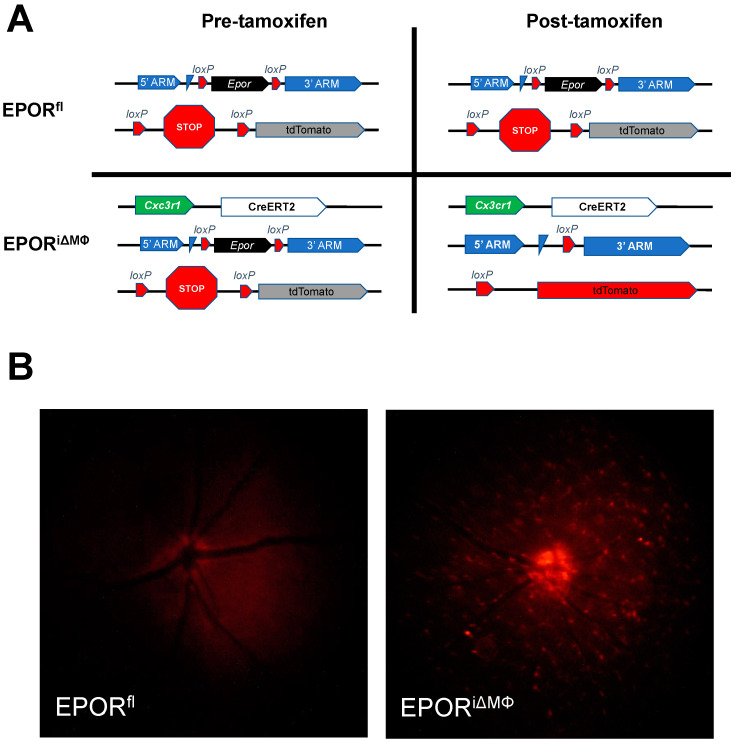
Validation of tamoxifen-inducible macrophage-specific Cre-loxP mouse model. (**A**) Schematic of Cre-mediated recombination in control mice (EPOR^fl^) that lack Cx3cr1-CreERT2 alleles (top row) or in experimental macrophage-specific knockout mice (EPOR^iΔMΦ^) that contain a Cx3cr1-CreERT2 allele (bottom row); (**B**) live fluorescence imaging using the Micron IV that demonstrated no tdTomato reporter expression in Cre null EPOR^fl^ (left), but tdTomato reporter expression in EPOR^iΔMΦ^ (right) two-weeks after tamoxifen administration (note autofluorescence of the optic nerve in the EPOR^fl^).

**Figure 3 biomedicines-10-01655-f003:**
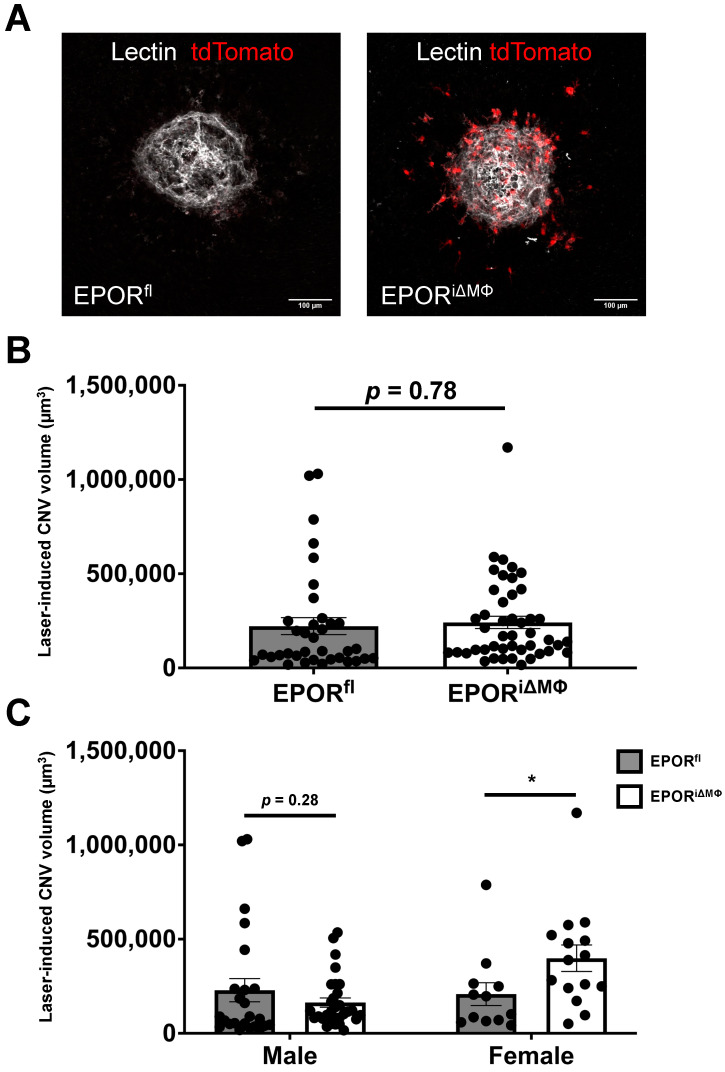
EPOR knockout in macrophages did not affect laser-induced CNV. (**A**) Representative confocal images of laser-induced CNV lesions from EPOR^fl^ (left) and littermate EPOR^iΔMΦ^ (right) that demonstrate lectin (white) and tdTomato (red) staining; (**B**) quantification of laser-induced CNV volumes (mean ± SEM, *n* = 36 lesions from EPOR^fl^ and *n* = 45 lesions from EPOR^iΔMΦ^); (**C**) quantification of laser-induced CNV volumes stratified by sex (mean ± SEM, * *p* < 0.05, *n* = 24 lesions from male EPOR^fl^, 30 lesions from male EPOR^iΔMΦ^, 12 lesions from female EPOR^fl^, and 15 lesions from female EPOR^iΔMΦ^).

**Figure 4 biomedicines-10-01655-f004:**
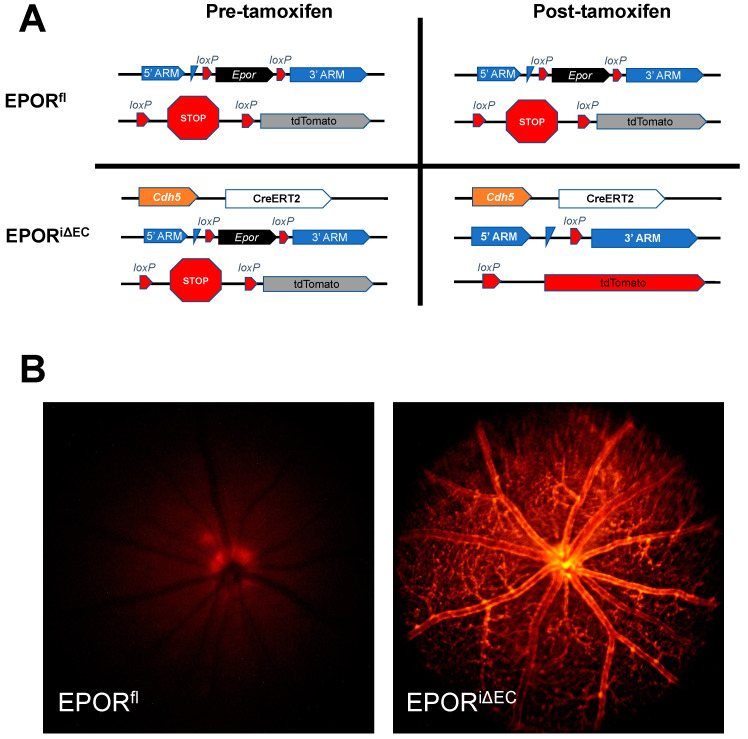
Validation of tamoxifen-inducible endothelial-specific Cre-loxP mouse model. (**A**) Schematic of Cre-mediated recombination in control mice (EPOR^fl^) that lack Cdh5-CreERT2 alleles (top row) or in experimental endothelial cell-specific knockout mice (EPOR^iΔEC^) that contain a Cdh5-CreERT2 allele (bottom row); (**B**) live fluorescence imaging using the Micron IV that demonstrated no tdTomato reporter expression in Cre null EPOR^fl^ (left), but tdTomato reporter expression in EPOR^iΔEC^ (right) two-weeks after tamoxifen administration (note autofluorescence of the optic nerve in the EPOR^fl^).

**Figure 5 biomedicines-10-01655-f005:**
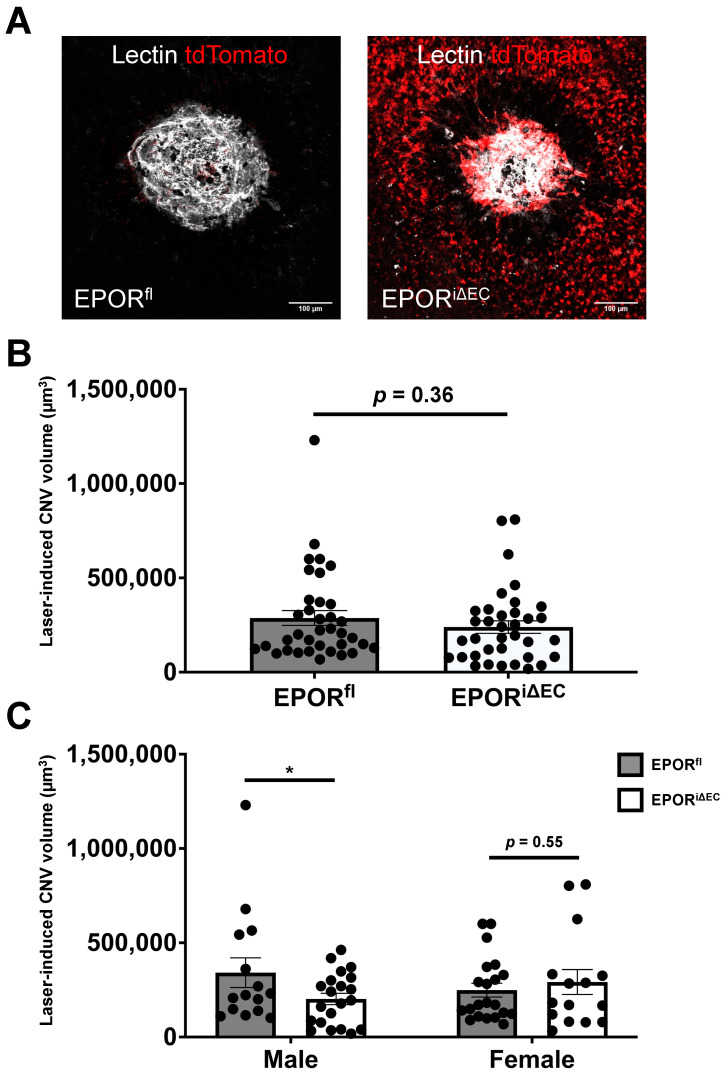
EPOR knockout in endothelial cells reduced laser-induced CNV in male mice. (**A**) Representative confocal images of laser-induced CNV lesions from EPOR^fl^ (left) and littermate EPOR^iΔEC^ (right) that demonstrate lectin (white) and tdTomato (red) staining; (**B**) quantification of laser-induced CNV volumes (mean ± SEM, *n* = 39 lesions from EPOR^fl^ and *n* = 39 lesions from EPOR^iΔEC^); (**C**) quantification of laser-induced CNV volumes stratified by sex (mean ± SEM, * *p* < 0.05, *n* = 15 lesions from male EPOR^fl^, 24 lesions from male EPOR^iEC^, 24 lesions from female EPOR^fl^, and 15 lesions from female EPOR^iΔEC^).

**Figure 6 biomedicines-10-01655-f006:**
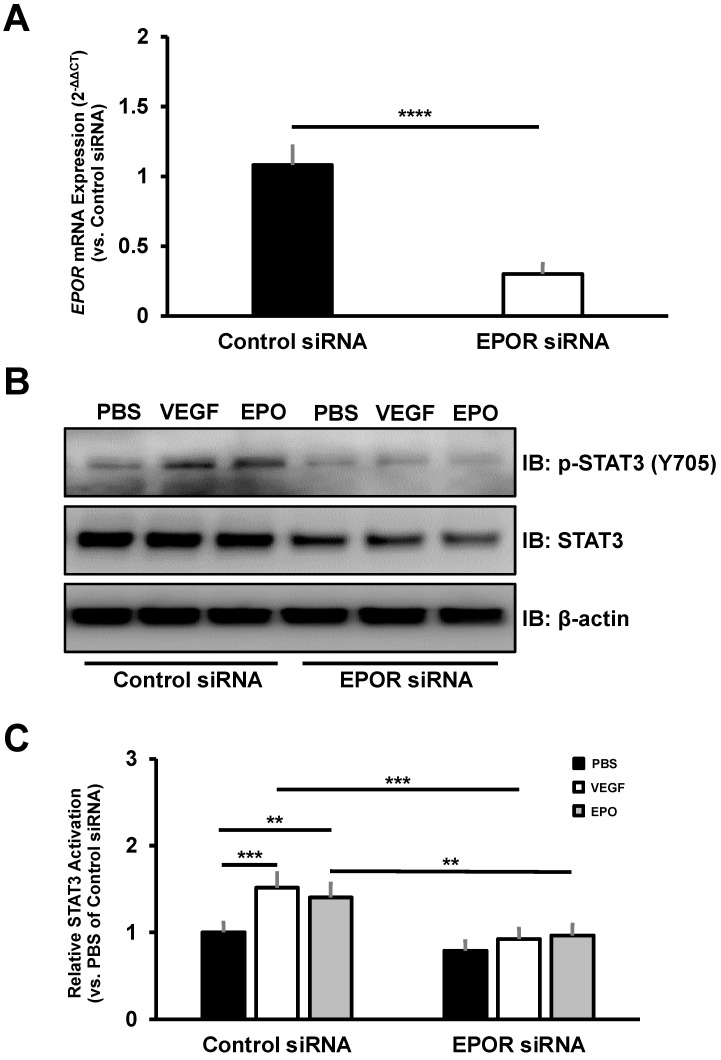
EPOR-mediated signaling is required for STAT3 activation in cultured CECs. (**A**) Sufficient knockdown of *EPOR* mRNA was observed in cultured CECs (normalized mean ± SEM, **** *p* < 0.0001, *n* = 3 biologic replicates per group from 3 different donors); (**B**) Representative western blot images that depict immunoblotting of phosphorylated STAT3 at tyrosine 705 residue (p-STAT3 (Y705), top image), total STAT3 (middle image), or β-actin (bottom image) in lysates from CECs transfected with control siRNA (columns 1–3) or EPOR siRNA (columns 4–6) and treated with either PBS, VEGF, or EPO; (**C**) quantification of STAT3 activation by densitometry analysis (normalized mean ± SEM, ** *p* < 0.01, *** *p* < 0.001, *n* = 3 biologic replicates per group from 3 different donors).

## Data Availability

The data that support the findings of this study are available from the corresponding author, M.E.H., upon reasonable request.

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
