# Peer review of "Role of Erythropoietin Receptor Signaling in Macrophages or Choroidal Endothelial Cells in Choroidal Neovascularization"

_biomedicines, 2022, doi:10.3390/biomedicines10071655_

Round 1

Reviewer 1 Report

The paper by Ramshekar  and co-workers faces the understating of the role of EPO/PEOR signaling in macrophages and endothelial cells in choroids in relation to atrophic AMD and neovascularization. The paper considers the potential safety issues related to EPO treatment/systemic upregulation in atrophic AMD and its potential further development in CNV. Selective macrophage and endothelial cells EPOR gene inactivation produced differential results. While macrophages seem not involved, choroid endothelial cells play a role in the disease development.

The paper is interesting since the topic merits attention for effective drug development. 

Major concern:

I suggest to put attention to the last part of the introduction by writing the driving hypothesis of the experimental setting. In the present version, only macrophages are taken into consideration and not choroidal endothelial cells.

Reviewer 2 Report

Erythropoietin receptor (EPOR)-mediated signaling can have some neuroprotective effects in the retina. The main goal of the study is to examine the EPOR pathway in laser-induced choroidal neovascularization (CNV). In their previously published papers, the same group of authors reported the increased laser CNV size and macrophages infiltration in knock-in mice with humanized (EPOR). Mouse models of tamoxifen-induced Epor knockout in either monocytes/microglia or endothelial cells were used in the current manuscript. The results show that Cx3cr1-Cre-driven Epor knockout did not affect CNV development, while Knockout Epor in endothelial cells decreased lesion size of laser CNV in male mice. Collectively, these findings are important and suggest that EPO is unlikely to be a therapeutic option for used in patients with age-related macular degeneration. In fact, its pro-inflammatory or pro-angiogenic effects raise concerns on its use in any ocular conditions. Overall, the studies were well designed, the mouse models were validated and the data support the authors’ conclusions. A few minor comments are listed below.

1    Fig. 1A and 1B. It appeared that the EPOR mRNA levels were measured by TaqMan Gene Expression assay and normalized to house-keeping gene(s). The y-axis labels were “probe intensity”, which was not explained elsewhere in the manuscript. Also, the statistical analyses were performed with mixed effects linear regression model. Please provide some details on data analyses and interpretation, such as what continuous variables were used for the regression analyses.

2    Fig. S1A and S1B were not available for review

3   Fig. 6, siRNA knockdown on EPOR in cultured human choroidal endothelial cells reduced STAT3 activation by VEGF or EPO. As a control, please provide data to show the effects of siRNAs on any of the CEC marker proteins

Reviewer 3 Report

This work is a follow up study on the authors’ previous work where they found overactive EPO signaling increasing choroidal macrophage numbers and CNV volume. Here, by generating tamoxifen-inducible macrophage-specific or endothelial cell-specific EPOR knockout mice, the authors decipher whether it is EPOR signaling in macrophages or endothelial cells that is important for this effect. They first confirm EPOR mRNA expression in RPE/choroid tissue in human donor eyes, and using publicly available single cell RNA sequencing databases, observe EPOR mRNA expression in macrophages and choriocapillaris cells. Then using the tamoxifen inducible cell specific mice they found no differences in laser-induced CNV size in macrophage specific EPOR knockouts, however, in endothelial specific EPOR knockout mice, a reduction in CNV volume was observed, though only in male mice. Finally, knockdown of EPOR in cultured human choroidal endothelial cells reduced EPO- and VEGF-induced STAT3 activation. The experimental design is clear and logical, however, I have some concerns/questions:

·       Authors state the current study was not powered by sex, which is a pity. Results from males vs females are often showing opposite effects, e.g. Fig 3C. P value for females is 0.08 with low n numbers of 15 and 12 lesions compared to 24 lesions and 30 lesions in males. There is a high chance that the difference in females could become significant by adding more n numbers so that it is powered for the study. These results will certainly add to the understanding of EPOR signaling in the 2 cells types in males versus females.

·       Were any differences in macrophage recruitment to the CNV site observed between EPORfl and EPORiDEC mice?

·       In fig 5A, right image, red td Tomato staining is seen all around the CNV in the RPE region. Should the staining not be endothelial cell specific?

Specific remarks:

·       A scatter plot with bar should be used for figures 3B, 3C, 5B and 5C to see the individual data points.

·       Line 306 ‘non-significant ~20% reduction in CNV volume’  - result is not significant so should be written as no significant changes in CNV volume.

Reviewer 4 Report

Ramshekar et al. realized a very interesting article describing the “Role of erythropoietin receptor signaling in choroidal endothelial cells or macrophages in choroidal neovascularization”. I consider the manuscript very interesting but, at the same time, I suggest several revisions needed to improve the reliability and the completeness of the paper:

·         The “Introduction” section did not speak sufficiently about retinal compromission related to inflammation and vascular alterations, collateral targets of authors’ work. Thus, I suggest the authors to add more recent references related to inflammation, oxidative stress, and angiogenesis in association to retinal degenerations. The recent PMID: 34440511 and PMID: 34058230 could represent a substrate able to enforce the role of considered cellular mechanisms.

·         Are experiments realized at least in triplicate?

·         In 2.8 section the list of used primers for RT-PCR should be added.

·         In 2.10 section, the statistical analysis lacks post-hoc correction test (e.g. Bonferroni).

·         The Fig. 1 should be improved for resolution.

·         Finally, manuscript requires English revisions and typos correction.

Round 2

Reviewer 4 Report

The manuscript can be accepted in the present form